# CODIFI (Concordance in Diabetic Foot Ulcer Infection): a cross-sectional study of wound swab versus tissue sampling in infected diabetic foot ulcers in England

Andrea Nelson,[1] Alexandra Wright-Hughes,[2] Michael Ross Backhouse,[3] Benjamin A Lipsky,[4] Jane Nixon,[2] Moninder S Bhogal,[5] Catherine Reynolds,[2] Sarah Brown,[2] on behalf of the CODIFI collaborators

[1]School of Healthcare, University of Leeds, Leeds, UK
[2]Clinical Trials Research Unit, University of Leeds, Leeds, UK
[3]Leeds Institute of Rheumatic and Musculoskeletal Medicine, University of Leeds, Leeds, UK
[4]Division of Medical Sciences, University of Oxford, Oxford, UK
[5]School of Biomedical Sciences, University of Leeds, Leeds, UK

**Correspondence to**
Professor Andrea Nelson;
E.A.Nelson@leeds.ac.uk

## ABSTRACT

**Objective** To determine the extent of agreement and patterns of disagreement between wound swab and tissue samples in patients with an infected diabetic foot ulcer (DFU).

**Design** Multicentre, prospective, cross-sectional study.

**Setting** Primary and secondary care foot ulcer/diabetic outpatient clinics and hospital wards across England.

**Participants** Inclusion criteria: consenting patients aged ≥18 years; diabetes mellitus; suspected infected DFU. Exclusion criteria: clinically inappropriate to take either sample.

**Interventions** Wound swab obtained using Levine's technique; tissue samples collected using a sterile dermal curette or scalpel.

**Outcome measures** Coprimary: reported presence, and number, of pathogens per sample; prevalence of resistance to antimicrobials among likely pathogens. Secondary: recommended change in antibiotic therapy based on blinded clinical review; adverse events; sampling costs.

**Results** 400 consenting patients (79% male) from 25 centres. Most prevalent reported pathogens were *Staphylococcus aureus* (43.8%), *Streptococcus* (16.7%) and other aerobic Gram-positive cocci (70.6%). At least one potential pathogen was reported from 70.1% of wound swab and 86.1% of tissue samples. Pathogen results differed between sampling methods in 58% of patients, with more pathogens and fewer contaminants reported from tissue specimens. The majority of pathogens were reported significantly more frequently in tissue than wound swab samples (P<0.01), with equal disagreement for *S. aureus* and *Pseudomonas aeruginosa*. Blinded clinicians more often recommended a change in antibiotic regimen based on tissue compared with wound swab results (increase of 8.9%, 95% CI 2.65% to 15.3%). Ulcer pain and bleeding occurred more often after tissue collection versus wound swabs (pain: 9.3%, 1.3%; bleeding: 6.8%, 1.5%, respectively).

**Conclusion** Reports of tissue samples more frequently identified pathogens, and less frequently identified non-pathogens compared with wound swab samples. Blinded clinicians more often recommended changes in antibiotic therapy based on tissue compared with wound swab

## Strengths and limitations of this study

► The first appropriately powered prospective study to assess agreement between these two methods of wound culture sampling.
► Investigates the relationship between baseline characteristics and agreement between the types of specimen using multivariable modelling.
► Included a substudy to investigate the potential clinical relevance of the different amount of information gleaned from tissue and wound swab results by seeking opinion of blinded clinicians on whether the microbiology results indicate a need to change antibiotic therapy.
► This pragmatic study defined pathogens based on those reported by the clinical microbiology laboratory, so it may not reflect all organisms/isolates identified.
► Tissue collection and sample culturing methods were not standardised across hospital laboratories.

specimens. Further research is needed to determine the effect of the additional information provided by tissue samples.

**Trial registration number** ISRCTN52608451.

## INTRODUCTION

Diabetes mellitus is now a worldwide pandemic, with the prevalence in the USA now exceeding 14%.[1] In persons with diabetes, foot complications, most commonly ulceration related to peripheral sensory and motor neuropathy and peripheral arterial disease,[2 3] occur in 15%–25% during their lifetime.[4 5] At presentation, over half of diabetic foot ulcers (DFU) are clinically infected[6] and foot infection precedes approximately 80% of non-traumatic lower limb amputations.[4 7 8]

Infection is a clinical diagnosis made using classification guidelines to help clinicians to determine infection severity.[9] Antibiotics are commonly initiated immediately (empirical treatment) and the results of samples collected for identification of wound pathogens and their sensitivities are then used to tailor the antibiotic regimen, avoiding unnecessarily broad-spectrum therapy and antibiotic resistance.[10–12] Accurate culture results depend on collecting samples of infected tissue that is less likely to be contaminated by colonising flora. Sterile swabs for culture are widely available, quick and easy to use and can be collected by most types of healthcare personnel. Unfortunately, wound swabs typically sample superficial flora, including colonisers or contaminants, and because of their construction (usually cotton wool) may fail to grow anaerobic or fastidious pathogens. Recognising these limitations, many clinical microbiology laboratories offer only minimal processing of wound swabs. Alternatively, specimens may be collected by obtaining tissue from the base of the wound; this requires slightly more skill and time, but may reveal more pathogens and be less susceptible to contamination with non-pathogens. Despite exhortations to obtain tissue rather than wound swab samples from most authoritative guidelines,[9 13 14] many clinicians default to the wound swab method. Our previous systematic review identified few studies comparing results of wound swabs and tissue samples,[15] and these had limitations including retrospective designs, inclusion of patients with various types of wounds, small cohorts and lack of contemporaneous sampling. Uncertainty has not been resolved in subsequent studies. One study[16] retrospectively reviewed 54 pairs of samples (from people with DFU but not all of whom had a wound infection) and reported that wound swabs detected more species than tissue samples—finding additional species in 11.2% of cases, fewer species in 9.0% of cases and completely different organisms in 6.7%. In a second study, 50 patients with an infected DFU had both swab and tissue samples taken; with the latter considered the 'gold-standard', wound swabs had 100% sensitivity but <20% specificity.[17] A third study, which collected specimens from 56 patients with an infected DFU, noted that wound swabs missed organisms identified from tissue specimens, especially Gram-negative bacteria, in patients with more severe infections.[18]

A further limitation of the published literature is that investigators have made the assumption that tissue specimens are the 'gold-standard' for sampling, but this method may also miss wound flora. Hence, we proposed a study to assess agreement and extent of disagreement between the two methods of collecting wound specimens, by comparing the pathogens isolated from each method from the same wound.

# METHODS
## Study design
We assessed the agreement between culture results of tissue and wound swab samples in patients with a suspected infected DFU. We have published a detailed description of the study methods.[19]

This was a multicentre, cross-sectional study of 400 people with diabetes mellitus in English primary and secondary care foot ulcer/diabetic outpatient clinics and hospital wards. Foot ulcer infection was diagnosed clinically based on signs and symptoms using Infectious Diseases Society of America/International Working Group on the Diabetic Foot (IDSA/IWGDF) criteria; patients were eligible for enrolment if the clinician evaluating them planned to treat them with antibiotic therapy. Consenting patients had a wound swab and tissue sample taken from the same foot ulcer. These were processed and reported by the usual local clinical microbiology laboratory so that the information gathered would be relevant for clinical practice.

Coprimary endpoints were the extent of agreement between wound swab and tissue sampling for three microbiological parameters: (1) presence of isolates likely to be pathogens; (2) the number of bacterial pathogens reported per sample; and (3) the prevalence among likely pathogens of resistance to antimicrobials.

In addition, we investigated the clinical usefulness of the information provided by tissue versus wound swab samples using a blinded clinical review panel to interpret the microbiology results. Secondary objectives considered sampling-related adverse effects and the costs of sampling.

In a separate substudy, we investigated the clinical outcomes at 12 months after sampling and explored the prognostic factors related to ulcer healing.[20]

## Eligibility criteria
Patients were eligible if they had: a diagnosis of diabetes mellitus (type 1 or 2); were at least 18 years old; and had a suspected infected DFU (with or without bone infection, based on clinical signs and symptoms using IDSA/IWGDF criteria and the judgement of the investigator). Patients were excluded if: the treating clinician deemed it inappropriate to take a tissue or wound swab sample for any reason; the patient had previously been recruited into the study; or they were unwilling or unable to provide informed consent. Patients were not excluded if they were currently being, or had recently been, treated with antimicrobial therapy.

## Assessments
### Sample acquisition
We trained clinicians at all centres to collect samples using the UK Health Protection Agency standards,[21 22] which were subsequently updated,[23 24] via site visits, and an e-Learning package that we developed for this purpose.[25] After wound cleansing and debridement (if required), a physician, nurse or podiatrist first obtained the wound swab sample from the infected ulcer using Levine's technique.[26] A tissue sample was subsequently collected using a sterile dermal curette or scalpel and placed in the transport medium used locally. All samples were transferred to, and processed by, the centre's local clinical microbiology laboratory.[21–24] Study samples received no special labelling or processing.

## Clinical assessments

Baseline data included a medical history and examination, including for any signs or symptoms of wound infection, previous treatments, and classifying the current status of the foot ulcer using the Perfusion, Extent, Depth, Infection and Sensation (PEDIS) scale,[27] Wagner[28] and Clinical Signs and Symptoms Classification of Infection systems,[29] and investigators solicited level of pain in the ulcer immediately after each sample was obtained. Investigators reported adverse events associated with sample collection.

## Centre differences questionnaire

Each participating site, including its microbiology laboratory, completed a questionnaire regarding how they: acquired samples for culture; transported them to the laboratory; analysed the specimens; and reported the results to clinicians. We also requested that they report their local antibiotic protocols to allow evaluation of any potential differences among centres.

## Clinical panel review

We compared the proportion of patients for whom the antibiotic regimen actually prescribed by the attending medical team was 'appropriate', based on culture and sensitivity results of wound swab or tissue samples. We sent microbiology results, along with a record of the empirical antimicrobial regimen prescribed, for the first 250 recruited patients (three were subsequently excluded due to protocol deviation or incomplete review) to a panel of 13 senior clinicians who worked with a diabetic foot team and had antibiotic prescribing privileges. Each clinician received the results of cultures of patients' wound swab or tissue sample on different occasions, and was blinded to whether results were from a tissue or wound swab specimen, and if they were from the same or different patients. Clinicians were asked:

1. 'Are there any pathogens identified in the lab report that are not covered by the prescribed antimicrobial regimen? (Yes/No)'
2. 'If you answered 'yes' to question 1, would knowing this information lead you to prescribe an alternative antibiotic regimen for this patient? (Yes/No)'

## Sample size

Our sample size was based on the primary outcome of the reported 'presence or absence of a pathogen'. Our target sample size was 400, as we calculated that 399 patients would provide 80% power to detect a difference of ≥3% in the reported presence of a given pathogen, if overall prevalence was 10%, with 5% disagreement between the wound swab and tissue samples, using a two-sided McNemar's test at the 5% level of significance. This level of agreement would also provide a kappa statistic of 0.7. This calculation is based on lower prevalence organisms, such as *Pseudomonas aeruginosa*,[30] hence the power was higher for more prevalent species.

## Statistical analysis

All tests of statistical significance were two sided and based on results from the evaluable population, with P values and 95% CIs provided as appropriate.

The various microbiology laboratories reported pathogens at a range of taxonomic levels, which we grouped by a previously developed scheme designed to report statistics meaningfully, that is, by genus, species, and so on. For pathogens with a prevalence >8% we generated cross-tabulations of reported presence in wound swab and tissue: overall percentage prevalence; agreement and disagreement; unadjusted kappa for agreement; prevalence and bias-adjusted kappa for agreement; prevalence difference (tissue-wound swab, and 95% CI); and McNemar's test for differences. As the participating laboratories used a number of scales to quantify the extent of growth of a pathogen (eg, +/++/+++; +/++/+++/++++; scanty/light/moderate/heavy; scanty/+/++/+++; light, moderate, heavy), we derived these onto one 3-point scale reported as +/++/+++. We used the derived data to tabulate the extent of bacterial growth (none, + to +++) and calculate weighted kappa statistics.

We prespecified baseline factors to investigate their relevance in determining agreement between sample results, including: type of ulcer (ischaemic or neuroischaemic vs neuropathic); Wagner grade of ulcer (1–5); recent antimicrobial therapy; and wound duration. We generated an overall summary of pathogens,[31] and used univariable multinomial regression by centre to determine whether agreement was influenced by any of these factors.

Using univariable ordinal regression modelling we assessed the influence of baseline factors on the number of pathogens as follows: tissue sampling (compared with wound swab) had two or more extra pathogens reported; tissue sampling had one extra pathogen reported; tissue and wound swab sampling had the same number of pathogens reported; wound swab sampling had one or more extra pathogens reported. In both regression analyses, we included centre as a random effect and multiple imputation to impute missing baseline factors.

For the clinical panel study of appropriateness of antibiotic treatment we summarised whether the pathogens reported were, or were not, covered by the actual treating clinician's prescribed antimicrobial regimen. We also asked if, in the blinded clinician's opinion, a change in antibiotic therapy was required. We used McNemar's test to identify whether one sampling method identified more patients requiring a change in therapy than the other.

## RESULTS

### Recruitment

Between 15 November 2011 and 15 May 2013 we screened 680 patients, and enrolled 401 patients from 25 centres. We excluded one patient whose consent was lost and five for whom one or more sample was lost or misused, resulting in a full analysis set of 400 patients and an evaluable population of 395 patients (figure 1).

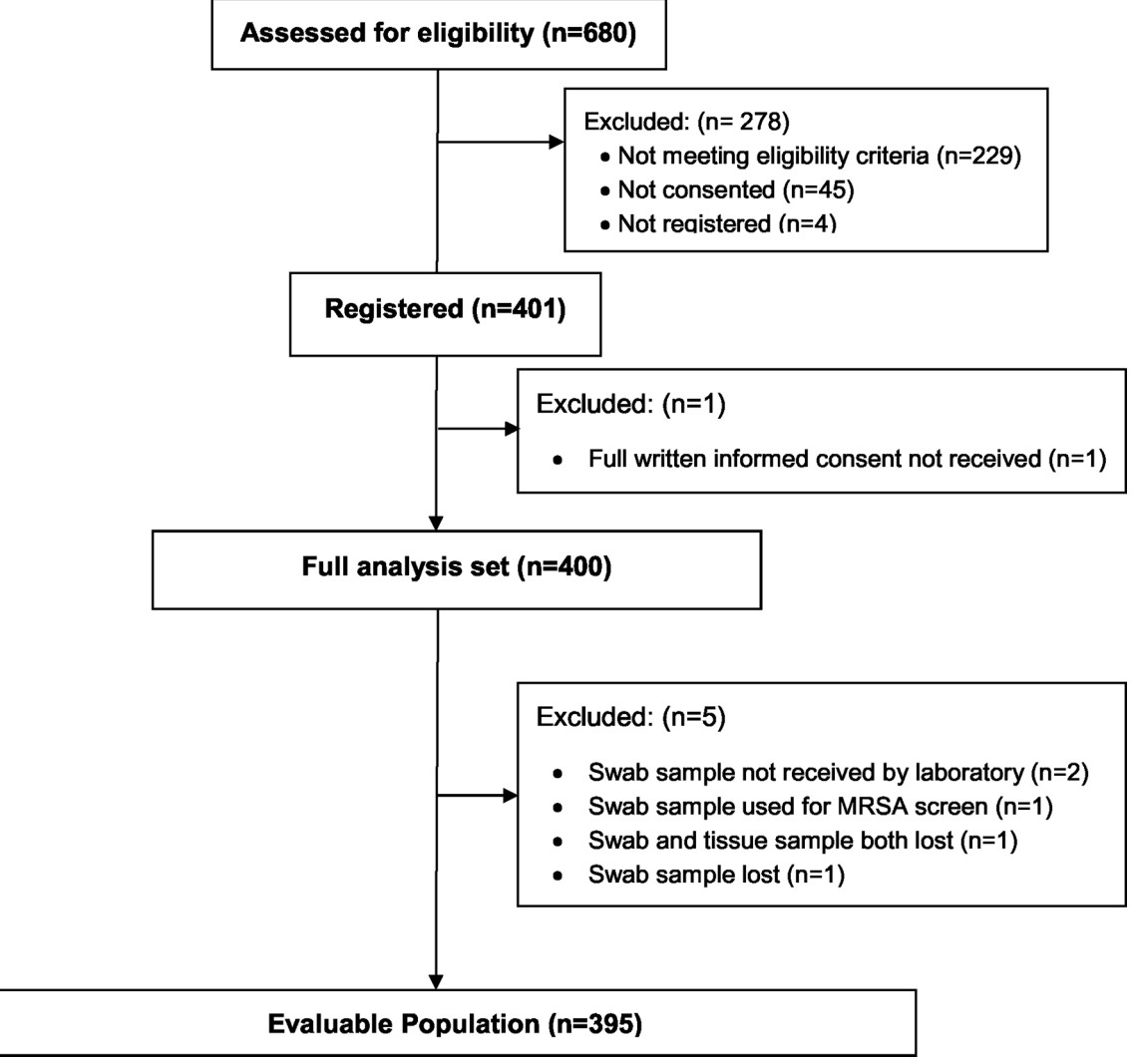

**Figure 1** Study recruitment diagram. MRSA, methicillin-resistant *Staphylococcus aureus*.

### Demographics

The recorded demographic characteristics of patients screened and those ultimately recruited were comparable. Most patients were recruited from outpatient clinics (79.8%) and were male (79.0%). Recruited patients had a median age of 63 years (range 26–99), a median duration of diabetes of 16.8 years (IQR 9–23) and median duration of their index ulcer of 5.6 months (IQR 0.7–6.0). Before sampling, 60.3% had an antimicrobial dressing or agent applied on the suspected infected ulcer, and 46.8% had received some type of systemic antibiotic therapy. After enrolment, 93.5% of patients received systemic antibiotic therapy (table 1).

### Microbiology results

Culture results yielded 79 different types of microbial isolates. Among the wound swab samples, there were no isolates reported from 20.0% and non-pathogenic isolates from 9.9%. Among tissue samples, there were no isolates reported in 10.1% and non-pathogenic isolates from 3.8% (table 2).

The most frequently reported groups of pathogens were: Gram-positive cocci (70.6%); Gram-negative bacilli (36.7%); *Enterobacteriaceae*, including coliforms (26.6%); obligate anaerobes (23.8%); and Gram-positive bacilli (11.1%). The most frequently reported pathogens were: *Staphylococcus aureus* (43.8%, of which 8.1% were methicillin resistant); *Streptococcus* (16.7%); *Enterococcus* (14.9%); coagulase-negative *Staphylococcus* (12.2%); *Corynebacterium* (9.4%); and *P. aeruginosa* (8.6%). All other genus and species level pathogens had a combined prevalence <6% (table 2).

### Primary endpoints
#### Summary of pathogens reported

For 58.0% of patients there was a difference in the pathogens reported by the two sampling techniques. The wound swab reported additional pathogens to those in the tissue sample in 8.1%; the tissue sample reported additional pathogens to those in the wound swab in 36.7%; and the tissue and wound swab samples reported different pathogens, with or without overlap, in 13.2%.

**Table 1** Baseline characteristics of enrolled patients

| Characteristic | Clinical values | Full analysis set (n=400) |
|---|---|---|
| Age (years) | Mean (SD) | 63.1 (13.3) |
| | Median, [range] and (IQR) | 63.0 [26-99] (54.0, 73.0) |
| Sex | Male | 316 (79.0%) |
| | Female | 84 (21.0%) |
| Ethnicity | White | 377 (94.3%) |
| | Other | 23 (5.7%) |
| Site of recruitment | Hospital ward | 53 (13.3%) |
| | Outpatient clinic | 319 (79.8%) |
| | Community clinic | 28 (7.0%) |
| Diabetes type | Type 1 | 58 (14.5%) |
| | Type 2 | 342 (85.5%) |
| Duration of diabetes (years) | n Missing | 3 |
| | Mean (SD) | 16.8 (11.0) |
| | Median, [range] and (IQR) | 15.0 [0.04–57] (9.0, 23.0) |
| Diabetes treatment details | Oral hypoglycaemic agent | 107 (27.8%) |
| | Insulin | 168 (43.6%) |
| | Oral hypoglycaemic agent and insulin | 109 (28.3%) |
| | Other | 1 (0.3%) |
| | None | 15 (3.8%) |
| Number of foot ulcers | 1 | 268 (67.0%) |
| | ≥2 | 132 (33.0%) |
| Duration of index ulcer (months) | n Missing | 4 |
| | Mean (SD) | 5.58 (12.28) |
| | Median,[range] and (IQR) | 1.84 [0.1–144.0] (0.69, 6.00) |
| Aetiology of index ulcer | Ischaemic | 14 (3.5%) |
| | Neuropathic | 202 (50.5%) |
| | Ischaemic and neuropathic | 182 (45.5%) |
| | Missing | 2 (0.5%) |
| Antimicrobial dressing on ulcer | Yes | 241 (60.3%) |
| | No | 154 (38.5%) |
| | Missing | 5 (1.3%) |
| Patient already on systemic antibiotics | Yes | 187 (46.8%) |
| | No | 194 (48.5%) |
| | Missing | 19 (4.8%) |
| Patient on antibiotics immediately after sampling | Yes | 374 (93.5%) |
| | No | 26 (6.5%) |

Continued

**Table 1** Continued

| Characteristic | Clinical values | Full analysis set (n=400) |
|---|---|---|
| Grade (Wagner scale)* | Grade 1 | 136 (34.0%) |
| | Grade 2 | 134 (33.5%) |
| | Grade 3 | 122 (30.5%) |
| | Grade 4 | 7 (1.8%) |
| | Grade 5 | 1 (0.3%) |

*Grade 1—superficial diabetic ulcer (partial or full thickness); grade 2—ulcer extension ligament, tendon, joint capsule, or deep fascia without abscess or osteomyelitis; grade 3—deep ulcer with abscess, osteomyelitis or joint sepsis; grade 4—gangrene localised to portion of forefoot or heel; grade 5—extensive gangrenous involvement of the entire foot.

### Reported presence of pathogens

The majority of pathogens were reported significantly more frequently in the tissue than the wound swab samples (P<0.01). For isolates of *S. aureus* and *P. aeruginosa,* however, there was equal disagreement, meaning that for the same number of patients wound swabbing missed a pathogen reported by tissue sampling, as there were pathogens missed by tissue sampling but reported by wound swabbing. A full cross-tabulation of the reported presence of all of these pathogens is shown in table 2, with statistical analyses presented in table 3.

We examined whether the outcome 'both wound swab and tissue report the same pathogens' was related to any of several potentially important patient baseline variables (table 4). Based on a summary of our results we performed a univariable multinomial analysis and found that none of the baseline factors examined had a significant effect on overall agreement.

### Reported presence of antimicrobial resistance among likely pathogens

We investigated the reported presence of three common antimicrobial-resistant pathogens using two sampling methods. Methicillin-resistant *S. aureus* was reported in 6.8% of wound swabs and 7.8% of tissue samples, a difference of 1.0% (95% CI –0.2% to 2.8%, McNemar's exact P value=0.219). Vancomycin-resistant *Enterococcus* was reported in only one (0.3%) patient (detected by both wound swab and tissue).

### Number of pathogens reported per sample

Comparing the number of pathogens isolated from tissue versus wound swab specimens, both had a median of 1.0 pathogen per sample, but the means were 1.5 and 1.0 and the maximum numbers were 6 and 4 pathogens, respectively. A greater proportion of wound swab samples reported no pathogens compared with tissue samples (29.9% vs 13.9%, respectively). In terms of number of pathogens reported for the tissue versus the wound swab sample, for 49.6% of patients they were the same, for 41.5% there was at least one more pathogen reported

**Table 2** Cross-tabulation of reported presence of at least one pathogen and pathogens with >8% prevalence in order of taxonomic rank and prevalence

| Pathogen (overall prevalence) | | | Tissue results Not reported | Tissue results Reported | Total |
|---|---|---|---|---|---|
| At least one pathogen (88.1%) | Swab | Not reported | 47 (11.9%) | 71 (18.0%) | 118 (29.9%) |
| | Swab | Reported | 8 (2.0%) | 269 (68.1%) | 277 (70.1%) |
| | Swab | Total | 55 (13.9%) | 340 (86.1%) | 395 (100.0%) |
| Gram-positive cocci (70.6%) | Swab | Not reported | 116 (29.4%) | 68 (17.2%) | 184 (46.6%) |
| | Swab | Reported | 14 (3.5%) | 197 (49.9%) | 211 (53.4%) |
| | Swab | Total | 130 (32.9%) | 265 (67.1%) | 395 (100.0%) |
| Gram-negative bacilli (36.7%) | Swab | Not reported | 250 (63.3%) | 49 (12.4%) | 299 (75.7%) |
| | Swab | Reported | 12 (3.0%) | 84 (21.3%) | 96 (24.3%) |
| | Swab | Total | 262 (63.3%) | 133 (33.7%) | 395 (100.0%) |
| Enterobacteriaceae (including coliforms) (26.6%) | Swab | Not reported | 290 (73.4%) | 37 (9.4%) | 327 (82.8%) |
| | Swab | Reported | 14 (3.5%) | 54 (13.7%) | 68 (17.2%) |
| | Swab | Total | 304 (77.0%) | 91 (23.0%) | 395 (100.0%) |
| Obligate anaerobes (23.8%) | Swab | Not reported | 301 (76.2%) | 46 (11.6%) | 347 (87.8%) |
| | Swab | Reported | 19 (4.8%) | 29 (7.3%) | 48 (12.2%) |
| | Swab | Total | 320 (81.0%) | 75 (19.0%) | 395 (100.0%) |
| Gram-positive bacilli (11.1%) | Swab | Not reported | 351 (88.9%) | 40 (10.1%) | 391 (99.0%) |
| | Swab | Present | 1 (0.3%) | 3 (0.8%) | 4 (1.0%) |
| | Swab | Total | 352 (89.1%) | 43 (10.9%) | 395 (100.0%) |
| *Streptococcus* (16.7%) | Swab | Not reported | 329 (83.3%) | 18 (4.6%) | 347 (87.8%) |
| | Swab | Reported | 5 (1.3%) | 43 (10.9%) | 48 (12.2%) |
| | Swab | Total | 334 (84.6%) | 61 (15.4%) | 395 (100.0%) |
| *Enterococcus* (excluding VRE) (14.9%) | Swab | Not reported | 336 (85.1%) | 34 (8.6%) | 370 (93.7%) |
| | Swab | Reported | 6 (1.5%) | 19 (4.8%) | 25 (6.3%) |
| | Swab | Total | 342 (86.6%) | 53 (13.4%) | 395 (100.0%) |
| Coagulase-negative *Staphylococcus* (12.2%) | Swab | Not reported | 347 (87.8%) | 39 (9.9%) | 386 (97.7%) |
| | Swab | Reported | 1 (0.3%) | 8 (2.0%) | 9 (2.3%) |
| | Swab | Total | 348 (88.1%) | 47 (11.9%) | 395 (100.0%) |
| *Corynebacterium* (9.4%) | Swab | Not reported | 358 (90.6%) | 33 (8.4%) | 391 (99.0%) |
| | Swab | Reported | 1 (0.3%) | 3 (0.8%) | 4 (1.0%) |
| | Swab | Total | 359 (90.9%) | 36 (9.1%) | 395 (100.0%) |
| *Pseudomonas aeruginosa* (8.6%) | Swab | Not reported | 361 (91.4%) | 8 (2.0%) | 369 (93.4%) |
| | Swab | Reported | 8 (2.0%) | 18 (4.6%) | 26 (6.6%) |
| | Swab | Total | 369 (93.4%) | 26 (6.6%) | 395 (100.0%) |
| *Staphylococcus aureus* (excluding MRSA) (35.7%) | Swab | Not reported | 254 (64.3%) | 16 (4.1%) | 270 (68.4%) |
| | Swab | Reported | 16 (4.1%) | 109 (27.6%) | 125 (31.6%) |
| | Swab | Total | 270 (68.4%) | 125 (31.6%) | 395 (100.0%) |
| Methicillin-resistant *S. aureus* (8.1%) | Swab | Not reported | 363 (91.9%) | 5 (1.3%) | 368 (93.2%) |
| | Swab | Reported | 1 (0.3%) | 26 (6.6%) | 27 (6.8%) |
| | Swab | Total | 364 (92.2%) | 31 (7.8%) | 395 (100.0%) |

MRSA, methicillin-resistant *Staphylococcus aureus*; VRE, vancomycin-resistant *Enterococcus*.

from the tissue than the wound swab sample, and for 8.9% there was at least one more pathogen reported from the wound swab than the tissue sample.

By univariable ordinal analysis we found that patients' tissue samples were reported to have two or more additional pathogens significantly more often if their ulcer was present for ≥56 days than if it was present <56 days (OR 1.56, 95% CI 1.05 to 2.33, P=0.024).

### Clinical panel review

In 73.3% of the cases reviewed by the blinded panel there was moderate agreement on the requirement for a change

**Table 3** Summary of agreement and disagreement statistics for most prevalent pathogens and the report of at least one pathogen

| | Overall prevalence | Overall disagreement | Difference (95% CI)* | McNemar's P value | Overall agreement | Unadjusted kappa (95% CI) | PABAK |
|---|---|---|---|---|---|---|---|
| At least one pathogen | 88.1% | 20.0% | 15.9% (11.8% to 20.1%) | <0.0001 | 80.0% | 0.44 (0.34 to 0.53) | 0.60 |
| Gram-positive cocci | 70.6% | 20.8% | 13.7% (9.4% to 18.0%) | <0.0001 | 79.2% | 0.57 (0.50 to 0.65) | 0.58 |
| Gram-negative bacilli | 36.7% | 15.4% | 9.4% (5.6% to 13.1%) | <0.0001 | 84.6% | 0.63 (0.55 to 0.71) | 0.69 |
| Enterobacteriaceae (including coliforms) | 26.6% | 12.9% | 5.8% (2.3% to 9.3%) | 0.0013 | 87.1% | 0.60 (0.50 to 0.70) | 0.74 |
| Obligate anaerobes | 23.8% | 16.5% | 6.8% (2.9% to 10.8%) | 0.0008 | 83.5% | 0.38 (0.26 to 0.50) | 0.67 |
| Gram-positive bacilli | 11.1% | 10.4% | 9.9% (6.9% to 13.5%) | <0.0001† | 89.6% | 0.11 (−0.01 to 0.23) | 0.79 |
| *Streptococcus* | 16.7% | 5.8% | 3.3% (0.9% to 5.6%) | 0.0067 | 94.2% | 0.76 (0.66 to 0.85) | 0.88 |
| *Enterococcus* (excluding VRE) | 14.9% | 10.1% | 7.1% (4.0% to 10.1%) | <0.0001 | 89.9% | 0.44 (0.30 to 0.58) | 0.80 |
| Coagulase-negative *Staphylococcus* | 12.2% | 10.1% | 9.6% (6.7% to 12.9%) | <0.0001† | 89.9% | 0.26 (0.11 to 0.41) | 0.80 |
| *Corynebacterium* | 9.4% | 8.6% | 8.1% (5.4% to 11.2%) | <0.0001† | 91.4% | 0.13 (−0.01 to 0.28) | 0.83 |
| *Pseudomonas aeruginosa* | 8.6% | 4.1% | 0.0% (−2.0% to 2.0%) | 1.0000 | 95.9% | 0.67 (0.52 to 0.82) | 0.92 |
| *Staphylococcus aureus* (excluding MRSA) | 35.7% | 8.1% | 0.0% (−2.8% to 2.8%) | 1.0000 | 91.9% | 0.81 (0.75 to 0.87) | 0.84 |

*Tissue-swab.
†Exact P value/CI.
MRSA, methicillin-resistant *Staphylococcus aureus*; PABAK, prevalence and bias-adjusted kappa; VRE, vancomycin-resistant *Enterococcus*.

in therapy between the wound swab and the tissue samples (kappa 0.45, 95% CI 0.34 to 0.56). In 17.8% of cases the blinded clinician indicated that the tissue sample results would lead to a recommendation of change in therapy, while the wound swab sample would not indicate a need for change. In 8.9% of cases the blinded clinician indicated that the wound swab result would lead to a change in therapy whereas the tissue sample would not (increase of 8.9%, 95% CI 2.7% to 15.3%).

### Adverse events

Investigators reported 'bleeding of concern' during sample collection in 30 (7.6%) of the recruited patients; it was attributed to the wound swab in six patients (1.5%) and to tissue sampling in 27 patients (6.8%). Higher levels of pain after either wound swab or tissue sampling were reported by 42 (10.5%) of patients. Of these, five (1.3%) patients reported worse pain after wound swabbing compared with tissue sampling, and 37 (9.3%) patients reported worse pain after tissue sampling compared with wound swabbing.

### Centre differences

We received responses to our questionnaires from 22 centres. Regarding the tissue sampling technique, one site used a dermal curette to collect tissue samples and others used a scalpel. There were no differences in the amount of time for a wound swab and a tissue sample to reach the microbiology laboratory from the clinic and no difference in the time it took from the receipt of the samples to processing. Among responding centres, 4 of 17 (23.5%) reported slightly more urgent processing of tissue samples.

Microbiology laboratories performed a Gram-stained smear of the specimen more frequently for tissue than wound swab samples; of 19 laboratories, 9 (47.4%) did this for tissue only, 3 (15.8%) did it for both samples and 6 (31.6%) did not routinely perform Gram staining (but offered it on request in one laboratory). Of 18 laboratories, 10 (55.6%) reported all isolates grown from a tissue sample but tailored wound swab sample reports according to clinical details and likely microbiological significance of the isolates. Centre differences were apparent in the multinomial and ordinal regression analysis where its inclusion improved the fit of both models (P<0.001).

Because only two microbiology laboratories provided data on the cost of processing specimens, it was not possible for us to do an analysis by specimen type.

### DISCUSSION

To our knowledge, this is the largest comparison of the two main methods of sampling an infected DFU, the first to report detailed data on paired samples for each pathogen from paired samples and the first to examine the relationship between baseline characteristics and agreement between microbiology results by types of specimen using multivariable modelling.

We found that tissue sampling had a higher yield than wound swab specimens, hence providing more information on wound flora. While tissue sampling overall detected more organisms than wound swabs, both techniques missed some organisms. Thus, to some degree they provide complementary information and both techniques may be useful. The differences in the results

**Table 4** Multinomial and ordinal regression models for individually fitted baseline factors

| | | OR (95% CI) | AIC§ | Reduction in −2 Log L | df | P value |
|---|---|---|---|---|---|---|
| Multinomial summary of isolates | Both swab and tissue report the same pathogens. | | | | | |
| Null model | | | 941.29 | | | |
| Ulcer type* | | | 945.72 | 1.570 | 3 | 0.666 |
| Any ischaemic versus neuropathic only | Swab>pathogens compared with the tissue | 1.03 (0.48 to 2.20) | | | | |
| Any ischaemic versus neuropathic only | Tissue>pathogens compared with the swab | 0.86 (0.53 to 1.40) | | | | |
| Any ischaemic versus neuropathic only | Swab and tissue report totally different pathogens. | 0.68 (0.35 to 1.31) | | | | |
| Ulcer grade | | | 949.16 | 4.125 | 6 | 0.660 |
| Grade 2 versus grade 1 | Swab>pathogens compared with the tissue | 0.68 (0.26 to 1.78) | | | | |
| Grade 2 versus grade 1 | Tissue>pathogens compared with the swab | 1.08 (0.60 to 1.93) | | | | |
| Grade 2 versus grade 1 | Swab and tissue report totally different pathogens. | 1.14 (0.51 to 2.54) | | | | |
| Grade 3/4/5 versus grade 1 | Swab>pathogens compared with the tissue | 1.28 (0.52 to 3.11) | | | | |
| Grade 3/4/5 versus grade 1 | Tissue>pathogens compared with the swab | 1.60 (0.87 to 2.95) | | | | |
| Grade 3/4/5 versus grade 1 | Swab and tissue report totally different pathogens. | 1.55 (0.69 to 3.45) | | | | |
| Previous antibiotic therapy* | | | 946.28 | 1.005 | 3 | 0.800 |
| Yes versus no | Swab>pathogens compared with the tissue | 0.80 (0.36 to 1.80) | | | | |
| Yes versus no | Tissue>pathogens compared with the swab | 1.14 (0.69 to 1.89) | | | | |
| Yes versus no | Swab and tissue report totally different pathogens. | 1.10 (0.56 to 2.16) | | | | |
| Antimicrobial dressing* | | | 943.44 | 3.850 | 3 | 0.278 |
| Yes versus no | Swab>pathogens compared with the tissue | 1.13 (0.51 to 2.51) | | | | |
| Yes versus no | Tissue>pathogens compared with the swab | 0.69 (0.40 to 1.19) | | | | |
| Yes versus no | Swab and tissue report totally different pathogens. | 1.38 (0.66 to 2.89) | | | | |
| Wound duration (median split)* | | | 941.48 | 5.802 | 3 | 0.121 |
| <56 days vs ≥56 days | Swab>pathogens compared with the tissue | 0.94 (0.43 to 2.04) | | | | |
| <56 days vs ≥56 days | Tissue>pathogens compared with the swab | 1.75 (1.08 to 2.86)† | | | | |
| <56 days vs ≥56 days | Swab and tissue report totally different pathogens. | 1.14 (0.59 to 2.17) | | | | |
| Log wound duration (continuous)* | | | 944.97 | 2.318 | 3 | 0.509 |
| | Swab>pathogens compared with the tissue | 0.95 (0.72 to 1.25) | | | | |
| | Tissue>pathogens compared with the swab | 0.88 (0.74 to 1.04) | | | | |

**Table 4**   Continued

| | OR (95% CI) | AIC§ | Reduction in −2 Log L | df | P value |
|---|---|---|---|---|---|
| Swab and tissue report totally different pathogens. | 0.93 (0.74 to 1.18) | | | | |
| Ordinal summary of isolates | | | | | |
| Null model | | 917.72 | | | |
| Ulcer type*: any ischaemic versus neuropathic only | 0.90 (0.61 to 1.33) | 919.45 | 0.271 | 1 | 0.603 |
| Ulcer grade | | 920.16 | 1.559 | 2 | 0.459 |
| Grade 2 versus grade 1 | 1.33 (0.82 to 2.15) | | | | |
| Grade 3/4/5 versus grade 1 | 1.27 (0.78 to 2.07) | | | | |
| Previous antibiotic therapy*: yes versus no | 1.25 (0.81 to 1.91) | 918.56 | 1.154 | 1 | 0.283 |
| Antimicrobial dressing*: yes versus no | 0.76 (0.49 to 1.18) | 918.16 | 1.553 | 1 | 0.213 |
| Wound duration (median split)*: <56 days vs ≥56 days | 1.56 (1.05 to 2.33) | 914.62 | 5.097 | 1 | 0.024‡ |
| Log wound duration (continuous)* | 0.92 (0.80 to 1.05) | 918.15 | 1.571 | 1 | 0.210 |

Based on the evaluable population n=395.

*Factors with missing data from the 28 (7.1%) patients with at least one missing data item.

‡Significant at the 5% level.

§Smaller is better.

AIC, Akaike information criterion.

of the two sampling techniques may be related to: the tissue specimen providing a greater yield of organisms at collection; a lower rate of bacterial isolates dying during specimen transport; or differences in the way the microbiology laboratory handled or reported the culture results. In settings where obtaining specimens by wound swab remains the standard method, until we determine the clinical impact of choosing tissue over swab sampling, we suggest examining methods to increase the yield from wound cultures.

For chronic wounds, there is no gold standard method of diagnosing infection. The minority of samples in our study that reported no pathogens may reflect either a false positive diagnosis of infection[32] or a false negative culture related to the use of antimicrobial dressings and antibiotics prior to sampling. Alternatively, this finding may be related to: improper sampling technique (eg, not sufficiently expressing tissue fluid in Levine's technique[26]); transport media that fail to maintain the viability of wound swab pathogens; or a decision by the microbiology laboratory to report only pathogens that they deemed clinically significant.

A key clinical issue is how much, and what type of, information on ulcer flora is useful for clinicians managing patients with an infected DFU. While clinicians want to optimally target their antibiotic therapy, providing microbiology reports listing many organisms, including likely non-pathogenic or unusual isolates present in low numbers, may confuse rather than aid decision-making.

We do not know, based on our results or the available literature, if antibiotic treatment based on a more detailed microbiogram helps select an antimicrobial regimen that increases the likelihood of, or time to, resolution of infection, or the prevention of treatment-associated antibiotic resistance.

We found that when blinded clinicians were presented with tissue, as opposed to wound swab microbiology reports, they were more likely to recommend a change in antibiotic therapy. This suggests that the additional information tissue specimens provide could lead to more tailored antimicrobial regimens. We do not know, however, if this theoretical finding would be confirmed in clinical practice.

It is certainly important to adequately cover all likely pathogens in a potentially limb-threatening problem like diabetic foot infection. However, given the global emergency associated with antibiotic resistance related to overuse of this precious resource, we are cautious about recommending a wholesale change to adoption of tissue sampling as theoretically this is a technique that may lead to unnecessarily broad-spectrum prescribing. Furthermore, the bacterial flora in the wound at the time of sampling may differ from those present days later after empirical antibiotic therapy, when culture results are reported, potentially reducing the utility of this information.

This study has several strengths. We provided all centres with training on appropriate techniques for wound swab

and tissue sampling in an effort to minimise between-sample, and between-centre, differences. We prospectively enrolled a large number of patients at many clinical sites using a carefully defined protocol that required obtaining contemporaneous dual specimens on each patient. The study also has high external validity; as we had minimal exclusion criteria, we recruited patients in usual practice settings, members of the attending clinical teams obtained the samples and the local laboratories processed the specimens.

There were, of course, some potential weaknesses of the study. There were differences among laboratories in tissue collection and sample culturing methods. These differences reflect the pragmatic nature of the study and ensure the results are generalisable to National Health Service centres and laboratories across England. Furthermore, only a small minority of patients (7%) were recruited from primary care (as opposed to specialty clinic or inpatient) centres. This limited our ability to investigate whether there was any difference in the extent of agreement in the reporting of pathogens between primary and secondary care sites.

Previous reports comparing wound swab with tissue specimens have been small, single-centre studies, and produced mixed results. One retrospective study of 89 concomitantly obtained pairs of samples from 54 patients with DFUs (87% clinically infected)[16] found that culture results of superficial wound swabs did not correlate well with those obtained from deep tissue, but they summarised their results in terms of predictive value for infection, for which there is no good evidence (deep tissue samples are an imperfect gold standard for diagnosing infection). Another study of 50 patients with a DFU ulcer[17] that compared culture results of tissue against wound swab specimens found that reports agreed in only 50% of patients. In another study of 56 patients with diabetic foot infection, grouped according to the PEDIS grading system,[18] wound swab culturing identified all microorganisms isolated from the corresponding deep tissue culture in 90% of grade 2 wounds, and in 41.4% and 41.2% for grade 3 and 4 wounds, respectively.

We believe our results demonstrate the increased yield from tissue compared with wound swab specimens; the maximum information would be available when reports from both samples are obtained. Combined with the currently available literature, this reinforces the recommendations that tissue samples are preferred over swab specimens if one method is to be selected. However, current guidelines do not recognise the complementarity of information when both methods are used. What is still needed is further research on whether this increased information from tissue sampling results in more appropriate prescribing or better resolution of infection or improved wound healing. Furthermore, we need more research on whether molecular approaches that provide extended views of the microbiome in conjunction with new developments in near-patient testing improve clinical outcomes and antibiotic stewardship. Results of these further studies would inform the most appropriate method of obtaining specimens from DFUs.

**Acknowledgements** We thank the members of the Study Steering Committee (Professor J Deeks, Professor R Cooper, Professor R Gadsby, Professor AM Keenan, Mrs C Thomas).

**Collaborators** The CODIFI collaborators were A Nelson, J Nixon, S Brown, J Gray, J Firth, C Dowson, E Jude, T Dickie, C Amery, G Sykes, P Vowden and M Edmonds.

**Contributors** AN (Professor of Wound Healing) was the chief investigator of the study and is the guarantor. She initiated the study, led the grant application development and the study team, and was involved in drafting and revising the paper critically for intellectual content. AWH (Senior Statistician) was a member of the study team, contributed to the drafting of the statistical analysis plan, undertook the statistical analyses, drafted the statistical results and contributed to the drafting and revising of the paper for intellectual content. MRB (NIHR Research Fellow, Podiatrist) was the first clinical study coordinator, a member of the study team, initiated sites for recruitment, contributed to the drafting of the study report and revised the paper for intellectual content. BAL (Professor of Medicine) was a coapplicant on the study grant, a member of the study team, contributed to the drafting of the paper and revised the paper for intellectual content. JN (Professor of Tissue Viability) was a coapplicant on the study grant, a member of the study team, the lead for the Clinical Trials Unit activity and responsible for these aspects. She contributed to the drafting of the manuscript and reviewed it for intellectual content. MSB (Senior Trial Coordinator) was the trials unit study coordinator and a member of the study team. He was responsible for ethics and governance applications and coordinated the study team. He contributed to the drafting of the study report and revised the paper for intellectual content. CR (Senior Data Manager) was the data manager and a member of the study team, responsible for data quality, contributed to the drafting of the study report and revised the paper for intellectual content. SB (Principal Statistician) was the supervising statistician, a coapplicant on the study grant, a member of the study team, contributed to the drafting of the statistical analysis plan, oversaw the statistical analyses and preparation of the statistical results. She contributed to the drafting of the manuscript and reviewed the paper for intellectual content.

**Funding** This work was supported by the National Institute for Health Research (NIHR) Health Technology Assessment (HTA) programme (project number 09/75/01).

**Disclaimer** The views and opinions expressed herein are those of the authors and do not necessarily reflect those of the HTA programme, NIHR, NHS or the Department of Health.

**Competing interests** None declared.

**Patient consent** Obtained.

**Ethics approval** CODIFI received ethical approval from the Sheffield NRES Committee (Ref: 11/YH/0078) and each enrolling site obtained local ethical approval prior to commencing recruitment.

**Provenance and peer review** Not commissioned; externally peer reviewed.

**Data sharing statement** Requests for data should be made to the corresponding author.

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
