## [Reviewer comments · BMJ Open]

ARTICLE DETAILS

TITLE (PROVISIONAL)	CODIFI (Concordance In Diabetic Foot Ulcer Infection) - a cross-sectional study of wound swabbing and tissue sampling in infected diabetic foot ulcers in England
AUTHORS	Nelson, E. Andrea; Wright-Hughes, Alexandra; Backhouse, Michael; Lipsky, BA; Nixon, Jane; Bhogal, Moninder; Reynolds, Catherine; Brown, Sarah

VERSION 1 – REVIEW

REVIEWER	Edgar Peters VU University Medical Center
REVIEW RETURNED	11-Oct-2017

GENERAL COMMENTS	The authors have submitted a report on the results of the CODIFI study. It is an eloquently written study to the concordance of cultures in diabetic foot infections. The research objective seems quite simple, but several sub analysis make the statistics more complicated. The research setting is the field of diabetic foot infection. This area of medicine is in need of strong data to guide clinical decisions. The number of patients enrolled is large enough to draw adequate conclusions. The minimal set of exclusion criteria probably leads to a high external validity of the conclusions. Although these strong point are important, a weak point of the study is that the choice of sampling and choice of antibiotic were not related to outcome. Samples were only compared, not the outcomes of choosing management based on the sample technique. Introduction: Another paper that might be added in the introduction is a paper by Sotto and coworkers (Diabetologia 2010, PMID: 20571753), It discusses outcomes of change of sampling from superficial swabbing to deep cultures and tissue biopsies. Although the authors did not directly compare tissue biopsies and swab results, the data did suggest differences in cultured microbiological organisms and cost savings. In the Sotto paper, it was suggested that tissue sampling lead to lower yield than swab (and to culturing less resistant microorganisms). Perhaps the authors could elaborate further on that in the discussion. Study design There is an unusual high percentage of female subjects. Do the authors have an explanation for this observation?
--

Is there any difference in primary care versus secondary care? The pathogens might be different, personnel might be trained differently, and available time might differ. If you have these data, please add these, if not, you could add that you do not have the data.

A foot infection was diagnosed clinically. Please add the diagnostic criteria (PEDIS?).

Swab and tissue samples were from the same wound at the same time. How can someone distinguish if the complication e.g. bleeding were from one of the sampling methods? Please add how you could make the distinction or remove the number of complications per sampling method from the paper.

Assessments:

All samples were processed at a local lab. These lab will used different microbiological techniques. How did these methods differ? Could differences have changed outcome? Page 13 has some remarks related to this, but does not give specific details.

Why did the researchers only send only results of 247 samples to the panel of blinded physicians? Were these randomly picked (if not, this could be a source of bias). Please add the reason not all samples were sent.

Could you give some background information of the clinicians of the panel? E.g. ID specialists, surgeons, years of experience?

Were there any subjects with suspected osteomyelitis? Did these subjects have different outcomes compared with subjects with soft tissue infection?

Results:

The last screen was performed in 2013: why was there such a long period before submission?

Of all subjects, 60% had antimicrobial dressings, 47% had systemic antimicrobial therapy. Was antimicrobial systemic or topical therapy withheld a couple of days before sampling? Could you add this to the text, please? Not withholding antimicrobial therapy could have a large influence on outcome and might explain the 30% of infections without growth of pathogens.

94% of patients received systemic antimicrobial treatment after enrolment. It seems odd that in case of clinical infection, the other 6% of patients were not treated with antimicrobial therapy. Were these 6% perhaps not infected? Please add a statement where you explain why the 6% were not treated.

The authors identify a number of bacteria. Some of the identified bacteria are in no way considered pathogens in other diabetic foot centres (e.g. Enterococcus, coagulase negative Staphylococcus, Corynebacterium). Presence of such organisms could have a major impact on choice of antimicrobial by the reviewing committee. If you would not consider these pathogens, you do not change your antimicrobial regime to cover these. Could you specify why these bacteria are considered pathogens and how presence of these (non-virulent) bacteria contributed to the opinion of the reviewing committee?

What were the standard therapies? E.g. In centres where coagulase negative Staphylococcus is considered a pathogen, it would make sense to have vancomycin in the empiric regimen (which leads to a lower change that the evaluation committee changes therapy.

REVIEWER	Matthew Malone Liverpool Hospital, South Western Sydney Local Health District, Australia
REVIEW RETURNED	16-Oct-2017

GENERAL COMMENTS	Well done on achieving such a large sample size from multiple sites. This study will augment current guidelines that promote tissue culture as the most appropriate sampling method. The above manuscript is a detailed prospective study exploring the concordance between swab samples and tissue cultures for DFI. Many expert guidelines promote the use of tissue samples as being superior to swab cultures for identifying pathogens of infection. Some studies have previously explored this idea and reported outcomes on smaller sample sizes. The current manuscript presents a significant sample size using multi-centres to collect data. The outcomes of the study re-enforce current practices and represents a significant contribution given the sample. This study should solidify and augment current practices and once again highlight the importance of taking appropriate cultures. I have some minor comments which may help improve the reading of the article. Minor comments:  1. line 44 page 5. What is symmetrical disagreement with respect to reviewing isolates? Can you not just state disagreement between blinded clinicians? 2. Would you not use the short hand spp. after species such as Pseudomonas or Streptococcus knowing the actual potential species are limited to a few in this genera. For example Streptococcus spp. 3. Page 2 line 57 – do you mean antibiotic therapy instead of antibiotherapy? 4. Page 3 line1. Strengths and Weakness – Is it worth here disclosing that standard culture techniques provide limited views of the microbiome with recent molecular approaches providing extended profiles. 5. Page 3 line 48. I think this section needs a little re-working. The diagnosis of infection are often based on clinical signs such as the presence of erythema, heat etc.. with the severity of infection graded using IDSA guidelines 2012. Maybe just expanding this first sentence to say infection is a clinical diagnosis using classification guidelines to help guide clinicians to determine disease severity. ? 6. Page 3 line 53. You state that antibiotics are started immediately etc,, I think this sentence could also be re-worked. Readers of BMJ will be aware what empiric means so no need to put in brackets. I would just state that antibiotics are provided empirically – I would also state that we commonly target aerobic gram-positive cocci as the majority of data has indicated these as the most common pathogens of infection. You state that a culture detects wound flora – but actually you do a culture to detect the most likely pathogens of infection not so much the flora.
---

Then I would start to discuss culture techniques. – this sets the scene nicely in this order.

7. On page 4 line 7. I think this needs re-wording - You state that culture results depend on collecting samples of infected tissue that is not contaminated by colonisers. The problem is that even tissue samples will contain colonisers or contaminants in addition to more likely pathogens. What I think your trying to say here is that some culture techniques are more likely to return greater yields of colonisers or contaminants because you just swab a wound surface, whilst obtaining tissue obtains deeper tissue / invading pathogens etc..

8. You use swabs a lot through the text, but I think its more insightful to state wound swabs

9. Page 4 line 15. You say that tissue sampling may obtain tissue from the base of the wound. There is no may. Tissue sampling does obtain tissue.

10. Page 5 line 17. How was foot infection diagnosed / classified ? was it using IDSA guidelines or IWGDF

11. Page 6. Line 8. You open with – We trained clinicians at all centres. Grammar wise I think it should be re-phrased. All Clinicians were appropriately trained across centres in obtaining swab and tissue cultures. I don't have access to any supplementary material you have submitted – I am unsure if you have discussed who obtained tissue, can nurses do this without advanced practice for example? The gold standard way to obtain tissue is biopsy so you may need an explanation somewhere as to why this wasn't done and you obtained debridement material only. Also, I cannot see the technique used for tissue collection or what transport medium and time frames of transport and anything on lab based methods used for culture analysis – I understand this may vary from lab to lab but it maybe worth stating that each lab utilised their own protocols for analysis– is this in a supplemented material?

12. Page 9, line 10. Im not sure you really need to disclose most were from clincis – what clinics if so. Maybe here is worth stating x% were from community based foot clincis and x% from hospital based high risk foot services? Better to delineate all areas or just leave out?

Again your opening demographics are confusing to readers. Most are women, then you state a mean age, is this of women or the total group.

13. Page 9 line20. I note most received antimicrobials of some sort. This is inevitable for the population size you used. I think that this requires further discussion how or if this is likely to reflect culture results – whilst this maybe true for your study determining concordance this doesn't matter, but in reflecting the microbiota associated with infected wounds then it may.

14. Page 10 line 12. Again some grammar corrections required. Instead of saying for most pathogen the prevalence was higher in tissues it should be tissue samples revealed a greater proportion of likely pathogens.. What is symmetrical agreement/disagreement ??

	15. Page 10, line 56. Typically ranges of variables are reported as ranges or even IQR with or without standard deviations? 16. The discussion is not balanced and virtually the entire section describes the strengths of the study. Actually, there are a number of limitations which are created through uncontrolled variables which you report as well-defined protocol. For example, your tissue collection methods were not standardized and were probably performed by a wide array of clinicians with different skills sets using different techniques from obtaining dermal curette versus debridement material. This is not discussed. The discussion section needs to be re-worked and I think structured to maybe: What are the main findings of your study How does this change clinical practice Do you recommended anything as a result of your findings? Discuss balance review of strengths and weakness Your microbiology results what do they infer about DFI? Future changes - What about role of molecular versus standard culture
--	--

VERSION 1 – AUTHOR RESPONSE

Reviewer: 1

Reviewer Name: Edgar Peters

Institution and Country: VU University Medical Center Please state any competing interests: none declared

Please leave your comments for the authors below

The authors have submitted a report on the results of the CODIFI study. It is an eloquently written study to the concordance of cultures in diabetic foot infections. The research objective seems quite simple, but several sub analysis make the statistics more complicated. The research setting is the field of diabetic foot infection. This area of medicine is in need of strong data to guide clinical decisions. The number of patients enrolled is large enough to draw adequate conclusions. The minimal set of exclusion criteria probably leads to a high external validity of the conclusions. Although these strong point are important, a weak point of the study is that the choice of sampling and choice of antibiotic were not related to outcome. Samples were only compared, not the outcomes of choosing management based on the sample technique.

The reviewer correctly acknowledges that an assessment of the relationship between sampling method and choice of antibiotic with clinical outcome was not conducted as part of this study. Indeed answering this question would not be possible within a cross sectional study where clinicians have access to reports from both sampling techniques from each patient. In the Discussion section we mention that results from this study demonstrate an increased sensitivity in detecting organisms by tissue compared to swab specimens; but, what remains unclear is whether this increased information results in more appropriate prescribing or better outcomes, including resolution of infection and healing. Therefore, we acknowledge that further research is needed to determine the effect of the additional information provided by tissue samples.

Addressing this issue would require investigating the relationship between sampling method and clinical outcome in a trial where patients are randomly allocated to a sampling method, rather than in a cross-sectional design.

Introduction:

Another paper that might be added in the introduction is a paper by Sotto and coworkers (Diabetologia 2010, PMID: 20571753), It discusses outcomes of change of sampling from superficial swabbing to deep cultures and tissue biopsies. Although the authors did not directly compare tissue biopsies and swab results, the data did suggest differences in cultured microbiological organisms and cost savings.

In the Sotto paper, it was suggested that tissue sampling lead to lower yield than swab (and to culturing less resistant microorganisms). Perhaps the authors could elaborate further on that in the discussion.

Response: We are familiar with the paper mentioned by Sotto et al, but we have a somewhat different view of the results of this retrospective review of his institutions experience over time. The authors state that after 2004 they did more “deep swab technique, including sharp wound debridement and curettage” compared to “swab technique” before that date. We think it is difficult to know what to make in the yearly fall of the number of isolates/sample over the next 3 years.

Study design

There is an unusual high percentage of female subjects. Do the authors have an explanation for this observation?

Response: We thank the reviewer for raising this issue, which led to us finding a mistake in the manuscript, which we have corrected. We actually recruited a high proportion of male patients (79.0%); this fact now corresponds to the data presented in Table 1. This high proportion of male participants is representative of the patient population screened for enrolment, which was 73.1% male.

We have now included a sentence stating that the demographic characteristics of patients screened and recruited were comparable.

Comment: Is there any difference in primary care versus secondary care? The pathogens might be different, personnel might be trained differently, and available time might differ. If you have these data, please add these, if not, you could add that you do not have the data.

Response: Only 7.0% of patients were recruited from primary care centres (community clinics). Therefore, we have limited data to draw robust conclusions on the relationship between primary and secondary care on the extent of agreement between swab and tissue sampling methods.

Comment: A foot infection was diagnosed clinically. Please add the diagnostic criteria (PEDIS?). Swab and tissue samples were from the same wound at the same time. How can someone distinguish if the complication e.g. bleeding were from one of the sampling methods? Please add how you could make the distinction or remove the number of complications per sampling method from the paper.

Response: The inclusion criterion in the protocol stated that the patient must have a suspected ulcer infection, with or without bone involvement, based on clinical signs and symptoms using Infectious Diseases Society of America / International Working Group on the Diabetic Foot (IDSA / IWGDF) criteria and the judgement of the investigator.

All swabs were taken before tissue samples. Thus, any pain or bleeding that occurred before collecting the tissue sample was considered to result from the swab, while pain/bleeding after the tissue sample were considered to result from the tissue sample. Given that the majority of adverse events occurred after tissue rather than swab sampling, this assumption appears valid. Furthermore, the decision on causality related to the collection method was made by local investigators in accordance with the principles of good clinical practice.

Assessments:

All samples were processed at a local lab. These lab will used different microbiological techniques. How did these methods differ? Could differences have changed outcome? Page 13 has some remarks related to this, but does not give specific details.

Resonse: All laboratories provided services to the NHS and worked according to Health Protection Agency standards for microbiology laboratories.

Please note the summary of the reported differences in laboratory handling of swab and tissue samples that we reported in Centre differences on page 13/14 of the manuscript. There were differences across laboratories with respect to the time to processing; there was more urgent processing of tissue samples in 4/17 (23.5%) laboratories. There were also differences according to whether the laboratory performed a Gram-stained smear; 9/19 (47.4%) did this for tissue only; 3/19 (15.8%) did it for both samples; whilst 6/19 (31.6%) did not perform Gram-staining (available only on request in 1 laboratory). Furthermore, 10/18 (55.6%) laboratories reported all isolates from a tissue sample but tailored reporting of swab sample results according to clinical details and their view of the organisms' significance. Therefore, differences in laboratories likely did have an effect on the isolates reported from swab versus tissue samples. However, these differences are in line with usual practice, and in accordance with the pragmatic nature of this study. Furthermore, we adjusted for Centre (and therefore any laboratory differences) in all of our analyses.

Comment: Why did the researchers only send only results of 247 samples to the panel of blinded physicians? Were these randomly picked (if not, this could be a source of bias). Please add the reason not all samples were sent.

Response: Results from swab and tissue samples were sent separately to clinicians for the first 250 participants recruited to the study; three were subsequently excluded due to protocol deviation or incomplete review. Our decision to limit the number of samples sent to the blinded physicians was based on allowing the clinical panel review to be completed by the end of the study, whilst also not over burdening reviewers.

In more detail, the clinical panel review was conducted in two rounds, each containing different participant's samples. During each round, a mix of swab and tissue sample results from different participants were sent to each blinded reviewer. After we received their reviews, the reviewers were then sent the sample participants corresponding swab or tissue sample result (again blinded to sample type, and participant). This process was repeated in a second round to maximise the number of participants included in the review, whilst also containing further validation samples for 30 participants to assess inter-rater reliability, and 30 participants to assess inter-rater reliability. The clinical panel review therefore took place over a number of months and each blinded reviewer reviewed a total of 13 – 31 participants samples, or 26 – 62 sets of results overall.

Could you give some background information of the clinicians of the panel? E.g. ID specialists, surgeons, years of experience?

Response: Clinicians' of the blinded reviewer panel comprised the local Principal Investigators across the study centres, all of whom were senior clinicians in the diabetic foot team, and had antibiotic prescribing rights. Please find that details have been added to the manuscript.

Comment: Were there any subjects with suspected osteomyelitis? Did these subjects have different outcomes compared with subjects with soft tissue infection?

Response: We expect there were probably some patients with osteomyelitis (based on their Wagner classification) however we do not know how many. Therefore we can only state that a total of 130 (32.5%) participants were reported to have a Wagner grade 3 or above (Table 1). Wagner grade was included in analysis comparing grade 1 ulcers to both grade 2, and grade 3 or above ulcers, however no statistically significant differences in outcomes were observed (Table 4).

Results:

The last screen was performed in 2013: why was there such a long period before submission? During the study we were given further funding to allow us follow-up the recruited participants for one year. This moved the study end date to December 2014, following final data collection and analysis. Since then our team has published the longer term follow-up results (now cited in the manuscript) and prepare the manuscript for publication of the main study results.

Of all subjects, 60% had antimicrobial dressings, 47% had systemic antimicrobial therapy. Was antimicrobial systemic or topical therapy withheld a couple of days before sampling? Could you add this to the text, please? Not withholding antimicrobial therapy could have a large influence on outcome and might explain the 30% of infections without growth of pathogens.

Response: In order to reflect clinical practice, antimicrobial therapy was not withheld prior to collection of samples. Most studies that have provided information on the microbiology of diabetic foot infections have reported that a substantial percentage of enrolled patients were currently or recently receiving some antimicrobial treatment.

We have expanded on the point made in the Discussion as stated below:

"The minority of samples in our study that reported no pathogens may reflect either a false positive diagnosis of infection³⁰ or a false negative culture related to the use of antimicrobial dressings and antibiotics prior to sampling."

Comment: 94% of patients received systemic antimicrobial treatment after enrolment. It seems odd that in case of clinical infection, the other 6% of patients were not treated with antimicrobial therapy. Were these 6% perhaps not infected? Please add a statement where you explain why the 6% were not treated.

Response: The inclusion criteria specified that patients had a suspected ulcer infection with a clinical plan to treat the patient with antibiotics for their infected ulcer. The reason that 6% of patients were not treated with systemic antimicrobial therapy after enrolment is unknown. We could only speculate on why this occurred, but we do not think that would be appropriate or useful. We can say, however, there was no indication that the pathogens reported were different from these patients compared to the 94% of patients who did receive systemic antimicrobial therapy.

Comment: The authors identify a number of bacteria. Some of the identified bacteria are in no way considered pathogens in other diabetic foot centres (e.g. Enterococcus, coagulase negative Staphylococcus, Corynebacterium). Presence of such organisms could have a major impact on choice of antimicrobial by the reviewing committee. If you would not consider these pathogens, you do not change your antimicrobial regime to cover these. Could you specify why these bacteria are considered pathogens and how presence of these (non-virulent) bacteria contributed to the opinion of the reviewing committee?

Response: While the three genera of organisms the reviewer cites (Enterococcus, coagulase-negative Staphylococcus, Corynebacterium) are considered low-virulence, and may often be colonisers, they have certainly been grown from aseptically obtained deep tissue (including bone) samples, and have been shown to be frequently present in studies that have used molecular microbiology techniques. Therefore, for carefully collected specimens (after cleansing and debridement), we decided during the planning of this study that it was appropriate to consider these specimens as presumptive pathogens. If reported, they were counted in both swab and tissue specimens, so there would be no bias in interpretation when comparing specimens.

Comment: What were the standard therapies? E.g. In centres where coagulase negative Staphylococcus is considered a pathogen, it would make sense to have vancomycin in the empiric regimen (which leads to a lower change that the evaluation committee changes therapy).

Response: Details of differences between centres will be reported in a separate publication. As the blind clinical review panel reviewed swab and tissue samples from the same patient then we consider the antibiotic prescribing policies at each centre is of low importance.

Reviewer: 2

Reviewer Name: Matthew Malone

Institution and Country: Liverpool Hospital, South Western Sydney Local Health District, Australia

Please state any competing interests: None to declare

Please leave your comments for the authors below. Well done on achieving such a large sample size from multiple sites. This study will augment current guidelines that promote tissue culture as the most appropriate sampling method.

Reviewer 3:

The CODIFI study of swab versus tissue sampling in infected diabetic foot ulcers - (CODIFI: Concordance In Diabetic Foot Ulcer Infection)

The above manuscript is a detailed prospective study exploring the concordance between swab samples and tissue cultures for DFI. Many expert guidelines promote the use of tissue samples as being superior to swab cultures for identifying pathogens of infection. Some studies have previously explored this idea and reported outcomes on smaller sample sizes.

The current manuscript presents a significant sample size using multi-centres to collect data.

The outcomes of the study re-enforce current practices and represents a significant contribution given the sample. This study should solidify and augment current practices and once again highlight the importance of taking appropriate cultures.

I have some minor comments which may help improve the reading of the article.

Minor comments:

1. line 44 page 5. What is symmetrical disagreement with respect to reviewing isolates? Can you not just state disagreement between blinded clinicians?

Response: We used the term symmetrical disagreement to reflect that the same proportion of swab and tissue sample results did not report isolation of *S. aureus* and *Pseudomonas* when the other sample results did, rather than the tissue sample reporting the pathogen more frequently than the swab, as was the case for the other pathogens.

The sentence in the manuscript has been revised for clarity with the wording 'symmetrical disagreement' replaced with 'equal disagreement'.

The disagreement between blinded clinicians then refers to a different outcome (requirement for a change in therapy) and analysis.

2. Would you not use the short hand spp. after species such as *Pseudomonas* or *Streptococcus* knowing the actual potential species are limited to a few in this genera. For example *Streptococcus* spp.

Response: The short hand spp. has now been removed from the manuscript.

3. Page 2 line 57 – do you mean antibiotic therapy instead of antibiotherapy?
The term antibiotherapy has been replaced with the term antibiotic therapy.

4. Page 3 line1. Strengths and Weakness – Is it worth here disclosing that standard culture techniques provide limited views of the microbiome with recent molecular approaches providing extended profiles.

Response: A brief statement acknowledging that recent molecular approaches providing extended profiles of the microbiome is now incorporated into the Discussion in the manuscript.

5. Page 3 line 48. I think this section needs a little re-working. The diagnosis of infection are often based on clinical signs such as the presence of erythema, heat etc.. with the severity of infection graded using IDSA guidelines 2012. Maybe just expanding this first sentence to say infection is a clinical diagnosis using classification guidelines to help guide clinicians to determine disease severity.
?

Response: As recommended by the reviewer, the first sentence in this section in the manuscript has been expanded to say "infection is a clinical diagnosis made using classification guidelines to help guide clinicians to determine infection severity".

6. Page 3 line 53. You state that antibiotics are started immediately etc,,, I think this sentence could also be re-worked. Readers of BMJ will be aware what empiric means so no need to put in brackets. I would just state that antibiotics are provided empirically – I would also state that we commonly target aerobic gram-positive cocci as the majority of data has indicated these as the most common pathogens of infection. You state that a culture detects wound flora – but actually you do a culture to detect the most likely pathogens of infection not so much the flora. Then I would start to discuss culture techniques. – this sets the scene nicely in this order.

Response: We agree with several of the points the reviewer makes here. As BMJ Open is a general medical journal, rather than one aimed at infectious diseases or wound care specialists, we think that the parenthetical term empirical treatment is appropriate as correct understanding of this term is central to interpreting the study. We agree that the term “pathogens” is more appropriate than “flora” and have changed that sentence. The issue of likely flora in diabetic foot wounds is not the point of this paper, we would opt to defer on further discussion of this point.

We revised the sentence in the manuscript to state: “Infection is a clinical diagnosis made using classification guidelines to help clinicians to determine infection severity.⁹ Antibiotics are commonly initiated immediately (empiric treatment) and the results of samples collected for identification of wound pathogens and their sensitivities are then used to tailor the antibiotic regimen, avoiding unnecessarily broad-spectrum therapy and antibiotic resistance. 10-12”

7. On page 4 line 7. I think this needs re-wording - You state that culture results depend on collecting samples of infected tissue that is not contaminated by colonisers. The problem is that even tissue samples will contain colonisers or contaminants in addition to more likely pathogens. What I think your trying to say here is that some culture techniques are more likely to return greater yields of colonisers or contaminants because you just swab a wound surface, whilst obtaining tissue obtains deeper tissue / invading pathogens etc..

Response: We agree with the reviewer and have revised the manuscript to state that culture results depend on collecting samples of infected tissue that is less likely to be contaminated by colonising flora.

8. You use swabs a lot through the text, but I think its more insightful to state wound swabs

Response: The manuscript has been revised throughout to state “wound swabs”

9. Page 4 line 15. You say that tissue sampling may obtain tissue from the base of the wound. There is no may. Tissue sampling does obtain tissue.

Response: The manuscript has been revised to state that tissue sampling obtains tissue from the base of the wound.

10. Page 5 line 17. How was foot infection diagnosed / classified ? was it using IDSA guidelines or IWGDF

Response: Please refer to the response we have provided to the same comment made by Reviewer 1.

The inclusion criteria in the protocol stated that the patient has a suspected ulcer infection with or without bone infection, based on clinical signs and symptoms using Infectious Diseases Society of America / International Working Group on the Diabetic Foot (IDSA / IWGDF) criteria and the judgement of the investigator.

11. Page 6. Line 8. You open with – We trained clinicians at all centres. Grammar wise I think it should be re-phrased. All Clinicians were appropriately trained across centres in obtaining swab and tissue cultures. I don't have access to any supplementary material you have submitted – I am unsure if you have discussed who obtained tissue, can nurses do this without advanced practice for example?

The gold standard way to obtain tissue is biopsy so you may need an explanation somewhere as to why this wasn't done and you obtained debridement material only. Also, I cannot see the technique used for tissue collection or what transport medium and time frames of transport and anything on lab based methods used for culture analysis – I understand this may vary from lab to lab but it maybe worth stating that each lab utilised their own protocols for analysis– is this in a supplemented material?

Response: We acknowledge the reviewer's comment on the grammar and the recommendation to rephrase to a passive voice, however we feel it is more appropriate to retain the active voice. We did indeed send members of our team to each site to train the clinicians on how to obtain the specimens.

We have reported the method of sample collection full in our previously published protocol paper (Nelson EA, Backhouse MR, Bhogal MS, et al. Concordance in diabetic foot ulcer infection. *BMJ Open* 2013; 3(1)) which we provide for the reviewer below. We cite the protocol paper in the manuscript to signpost the reader to the methods of wound swab and tissue sampling.

"We trained clinicians at all centres to collect samples using the HPA standards as a minimum requirement. Training was initially delivered during the site initiation visit but staff were also able to access an e-learning package containing a video at any time throughout the study. It is not anticipated that this substantially altered current swabbing practice as this is a routine procedure with established patterns of practice from the HPA.

After wound cleansing (using sterile saline and gauze) and debridement (removal of necrotic tissue, foreign material, callus, undermining of the wound edge), a physician, nurse or podiatrist will obtain specimens for aerobic and anaerobic cultures by

- ▶ First, using a cotton-tipped swab rubbed over the wound surface to sample superficial wound fluid and tissue debris. The swab will be pressed with sufficient pressure on the wound bed to capture expressed wound fluid, and will be positioned deep in the ulcer to collect from likely infected areas.
- ▶ Immediately after the cotton swab has been collected, a tissue sample will be removed from the same area of the ulcer bed. This procedure will be done using sterile equipment (forceps, scalpel and scissors) and aseptic technique. It will involve the removal of a small piece of wound tissue at the base of the wound by scraping or scooping using a dermal curette or sterile scalpel blade.

Clinicians in the participating sites will participate in a study information session to update their technique for acquiring wound samples. Clinicians will also view an e-learning package that will be developed and issued to all sites, detailing study procedures. This will include video footage of correct methods of obtaining both types of samples."

Information on the level of the health care professional was not obtained as it was not deemed relevant.

Centres were requested to process their samples at local laboratories, in line with their standard clinical practice. Samples were not labelled as part of the studies and would be processed in line with local standard operating procedures. It was expected that all centres followed HPA guidelines when processing the samples; these guidelines are cited in the manuscript. In order to understand the differences in the collection and processing of wound swab and tissue samples between centres, a survey was conducted to obtain information including the techniques used for tissue collection, transport medium, time frames for transportation etc. The data from this survey is planned for a future paper.

As stated in the response to Reviewer 1, differences between laboratories reflected usual clinical practice and therefore the pragmatic nature of the study. Centre (and therefore laboratory differences) was adjusted for in all analysis.

We know of no published data supporting biopsy (as opposed to other methods of obtaining tissue for culture) as the “gold standard” for obtaining tissue samples. What we believe this is the first paper published on specimen collection from diabetic foot infections (Sapico FL, Witte JL, Canawati HN, Montgomerie JZ, Bessman AN. The infected foot of the diabetic patient: quantitative microbiology and analysis of clinical features. Rev Infect Dis. 1984 Mar-Apr;6 Suppl 1:S171-6) demonstrated that with multiple specimens from the same patient, results of curettage specimens correlated extremely well with those of deep tissue biopsy specimens. Since that time, most DFI studies that reported tissue culture results have used curettage specimens; these appear to be as valid as tissue biopsy, are easier to collect, and are associated with fewer potential complications.

12. Page 9, line 10. Im not sure you really need to disclose most were from clincis – what clinics if so. Maybe here is worth stating x% were from community based foot clincis and x% from hospital based high risk foot services? Better to delineate all areas or just leave out?
The type of clinic (i.e., outpatient) is now incorporated in the manuscript for clarification.

Again your opening demographics are confusing to readers. Most are women, then you state a mean age, is this of women or the total group.
Please refer to our response to Reviewer 1; we have amended the wording to make this clearer.

13. Page 9 line20. I note most received antimicrobials of some sort. This is inevitable for the population size you used. I think that this requires further discussion how or if this is likely to reflect culture results – whilst this maybe true for your study determining concordance this doesn't matter, but in reflecting the microbiota associated with infected wounds then it may.

Response: Please see our response on this issue to a previous reviewer. Most reports of culture results from patients with DFIs don't provide data on the percentage of patients who have received recent antimicrobial therapy, and certainly not the route of administration, as we have in this study. Given the pragmatic nature of the study, the results reflect the standard of care in the NHS in England. Furthermore, as the reviewer suggests, the previous use of antimicrobials should not affect the assessment of concordance of wound swab and tissue sampling.

14. Page 10 line 12. Again some grammar corrections required. Instead of saying for most pathogen the prevalence was higher in tissues it should be tissue samples revealed a greater proportion of likely pathogens.. What is symmetrical agreement/disagreement ??

Response: For the question relating to symmetrical agreement/disagreement we refer the reviewer to our earlier response under point 1. We have revised the manuscript to state the following: 'The majority of pathogens were reported significantly more frequently in tissue than swab samples ($p < 0.01$), whereas there was equal disagreement for isolates of *S. aureus* and *Pseudomonas*.'

15. Page 10, line 56. Typically ranges of variables are reported as ranges or even IQR with or without standard deviations?

Response: The authors, in consultation with our biostatisticians, believe that we have used the appropriate measures of variability throughout the manuscript.

16. The discussion is not balanced and virtually the entire section describes the strengths of the study. Actually, there are a number of limitations which are created through uncontrolled variables which you report as well-defined protocol. For example, your tissue collection methods were not standardized and were probably performed by a wide array of clinicians with different skills sets using different techniques from obtaining dermal curette versus debridement material. This is not discussed.

The discussion section needs to be reworked and I think structured to maybe:

What are the main findings of your study

How does this change clinical practice

Do you recommended anything as a result of your findings?

Discuss balance review of strengths and weakness

Your microbiology results what do they infer about DFI?

Future changes - What about role of molecular versus standard culture

Response: We have revised the Discussion to incorporate the potential weaknesses/limitations of the study suggested by the reviewer.

The structure is in line with the guidance set out in the "instructions for authors".

VERSION 2 – REVIEW

REVIEWER	Matthew Malone Liverpool Hospital, Sydney
REVIEW RETURNED	02-Dec-2017
GENERAL COMMENTS	The authors have done well to incorporate the majority of suggested changes from reviewers and I am happy these have in most parts been addressed.